Corrected: Author correction

# A mechanistic framework for auxin dependent *Arabidopsis* root hair elongation to low external phosphate

Rahul Bhosale [1,2], Jitender Giri [2,3], Bipin K. Pandey [2,3], Ricardo F.H. Giehl[4], Anja Hartmann[4], Richard Traini[1,2], Jekaterina Truskina[1,2,5], Nicola Leftley[1,2], Meredith Hanlon[6], Kamal Swarup[1,2], Afaf Rashed[1,2], Ute Voß[1,2], Jose Alonso[7], Anna Stepanova[7], Jeonga Yun[7], Karin Ljung [8], Kathleen M. Brown [6], Jonathan P. Lynch [1,2,6], Liam Dolan[9], Teva Vernoux [5], Anthony Bishopp[1,2], Darren Wells [1,2], Nicolaus von Wirén [4], Malcolm J. Bennett [1,2] & Ranjan Swarup[1,2]

Phosphate (P) is an essential macronutrient for plant growth. Roots employ adaptive mechanisms to forage for P in soil. Root hair elongation is particularly important since P is immobile. Here we report that auxin plays a critical role promoting root hair growth in *Arabidopsis* in response to low external P. Mutants disrupting auxin synthesis (*taa1*) and transport (*aux1*) attenuate the low P root hair response. Conversely, targeting *AUX1* expression in lateral root cap and epidermal cells rescues this low P response in *aux1*. Hence auxin transport from the root apex to differentiation zone promotes auxin-dependent hair response to low P. Low external P results in induction of root hair expressed auxin-inducible transcription factors ARF19, RSL2, and RSL4. Mutants lacking these genes disrupt the low P root hair response. We conclude auxin synthesis, transport and response pathway components play critical roles regulating this low P root adaptive response.

[1] Plant & Crop Sciences, School of Biosciences, University of Nottingham, Nottingham LE12 5RD, UK. [2] Centre for Plant Integrative Biology (CPIB), University of Nottingham, Nottingham LE12 5RD, UK. [3] National Institute of Plant Genome Research (NIPGR), New Delhi 110067, India. [4] Leibniz Institute of Plant Genetics and Crop Plant Research (IPK), D-06466 OT Gatersleben, Stadt Seeland, Germany. [5] Laboratoire Reproduction et Développement des Plantes, Univ Lyon, ENS de Lyon, UCB Lyon 1, CNRS, INRA, F-69342 Lyon, France. [6] Department of Plant Science, The Pennsylvania State University, 102 Tyson Building, University Park, PA 16802, USA. [7] Department of Plant and Microbial Biology, NC State University, Raleigh, NC 27695, USA. [8] Umeå Plant Science Centre, Department of Forest Genetics and Plant Physiology, Swedish University of Agricultural Sciences, SE-901 83 Umeå, Sweden. [9] Department of Plant Sciences, University of Oxford, Oxford OX1 3RB, UK. These authors contributed equally: Rahul Bhosale, Jitender Giri, Bipin K. Pandey. Correspondence and requests for materials should be addressed to M.J.B. (email: malcolm.bennett@nottingham.ac.uk) or to R.S. (email: ranjan.swarup@nottingham.ac.uk)

Phosphorus is an essential element for plant growth and development, which is absorbed in the form of phosphate (P). This macro-nutrient is extremely reactive and readily precipitates with Al, Fe, or Ca making P one of the most limiting nutrients in soil. Plants have evolved a number of mechanisms to acquire soil P. Key adaptations under P deficient conditions include exudation of organic acids and phosphatases to solubilize soil P, induction of high affinity phosphate transporters and reprogramming of root system architecture (RSA).

Root architectural changes under P stress include altered root to shoot ratio, increased root surface area and shallower root angle to improve foraging in the P rich top soil[1–5]. At the cellular level, plants adapt to low P by increasing root hair elongation and density[6–14]. Root hairs are formed by specialized epidermal cells called trichoblasts[15]. The distribution of trichoblasts is species dependent and in *Arabidopsis*, trichoblasts (or H cells) are arranged in cell files interspersed with files of non-root hair forming cells (atrichoblast or N cells)[16]. Molecular genetic studies have identified several key genes regulating root hair development[1,17,18]. Among them bHLH transcription factors ROOT HAIR DEFECTIVE 6 (RHD6) and ROOT HAIR DEFECTIVE 6-LIKE 2 (RSL2), and ROOT HAIR DEFECTIVE 6-LIKE 4 (RSL4) appear to be crucial for hair morphogenesis[16]. Whereas RHD6 is required for early stages of trichoblast development, *RSL4* appears to be the key gene regulating hair length[19–21]. RSL2 has been shown to be responsive to P deficiency[22]. Despite these impressive recent advances, the molecular mechanisms regulating root hair elongation under P stress are much less well understood[6].

We report that the promotion of *Arabidopsis* root hair elongation under low external P requires the concerted activities of the auxin synthesis, transport, and response pathway components. Mutants disrupting these distinct auxin pathways attenuate the ability of root tissues to promote hair elongation in response to low external P conditions. Reporter lines revealed that under low P conditions auxin levels are up-regulated at the root tip, whilst targeted expression studies demonstrate that auxin is mobilized via AUX1 to the root hair differentiation zone. Elevated auxin levels in trichoblasts trigger a gene expression cascade, mediated by transcription factors ARF19, RSL2, and RSL4 that promotes hair elongation. In parallel papers, we demonstrate that this auxin-dependent root hair response to low external P is highly conserved in the monocot model rice[23] and relies on TIR1 to promote hair elongation via elevated intracellular auxin and calcium signaling responses[24].

## Results

**IAA increases in *Arabidopsis* roots grown in low external P.** Increased root hair elongation is a highly conserved adaptive response to low external P in many plant species including the eudicot and monocot models *Arabidopsis* (Fig. 1a) and rice[23]. Earlier genetic evidence suggested that auxin may facilitate root hair elongation under low external P conditions[6]. We initially tested whether auxin levels were up-regulated under low external

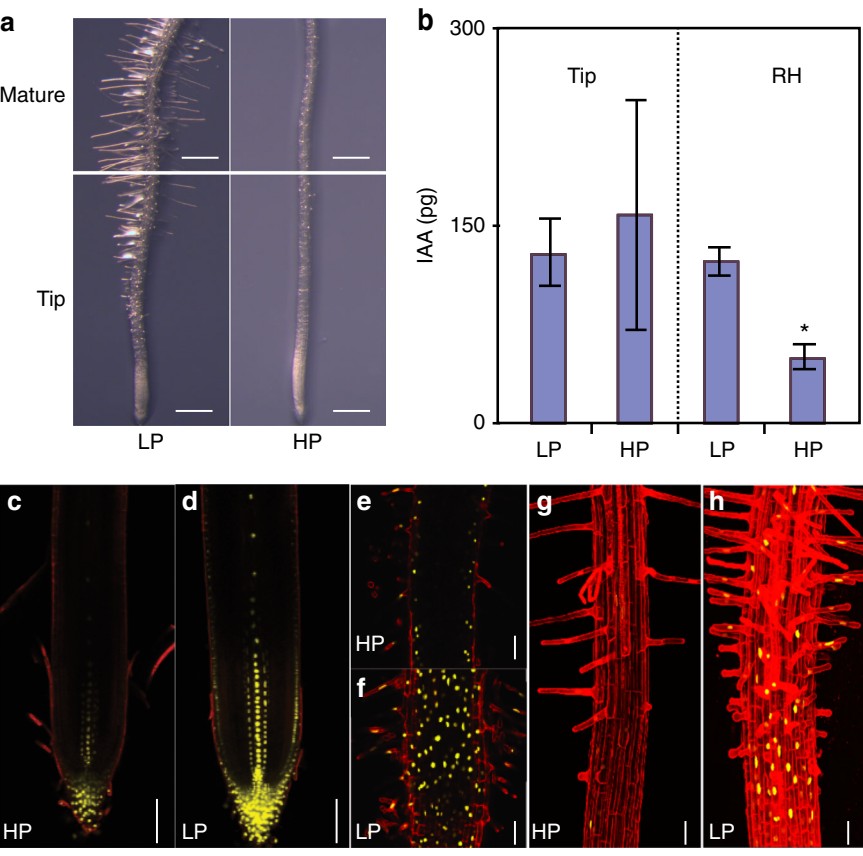

**Fig. 1** Root hair elongation is a conserved response to low P in *Arabidopsis* and results in IAA accumulation in the root tip. **a** Representative images of 9 day old seedlings grown in 3 μM (LP) or 312 μM P (HP). Scale bar = 0.5 mm. **b** IAA abundance was measured in 9 days old root tips (tip) and root hair region (RH) in *Arabidopsis* seedlings grown under low (LP) and high P (HP). Error bars represent standard error, N = 4. * Indicates significant difference (q-value < 0.05) judged by Student's *t*-test. **c–h**. Confocal images of 9 day old *Arabidopsis* roots expressing auxin response reporter *DR5:VENUS* (yellow) under low (LP) and high P (HP) showing median (**c**, **d**) and surface (**e**, **f**) views. **g** and **h** are maximum projections. Propidium iodide was used for counter staining (red). Scale bar = 50 μm

P conditions. *Arabidopsis* seedling root tips and root hair zone samples were micro-dissected, then their IAA levels were quantified using gas chromatography-tandem mass spectrometry (GC-MS/MS). Our MS profiling results revealed a significant increase in IAA accumulation in the root hair zone of seedlings grown under low P (3 μM) versus high P (312 μM) (Fig. 1b). We next tested whether the low P induced IAA levels resulted in an increase in auxin response. The DR5::VENUS auxin responsive reporter revealed an increased signal in the root apex (primarily in columella cells, quiescent center, stele, and lateral root cap cells) and root hair zone in seedlings grown under low P versus high P conditions (Fig. 1c–h). Our results reveal that *Arabidopsis* roots exposed to low external P conditions increase their IAA levels, triggering auxin responses in root tip and hair zone cells.

**Root hair elongation under low P requires auxin synthesis**. To probe the molecular events regulating root hair elongation under different external P conditions, we performed Agilent array-based gene expression profiling on seedling root tissues either grown under low or high P conditions (see Methods section and Gruber et al.[5]). Our array results revealed that transcript levels of the auxin biosynthesis gene *TRYPTOPHAN AMINOTRANSFERASE OF ARABIDOPSIS1, (TAA1)* increased in response to low external P conditions (Fig. 2a). This was validated using genomic *TAA1:GUS* translational reporter lines which revealed levels of the enzyme were up-regulated in root apices grown in low external P conditions (Supplementary Fig. 1). Array profiling also revealed that transcripts of the auxin inducible IAA degrading enzyme *DIOXYGENASE FOR AUXIN OXIDATION1* (*DAO1*) are significantly upregulated under low P conditions (Fig. 2a). Our transcriptomic results suggest that the regulation of key auxin homeostasis genes may play an important regulatory role during root hair elongation under low P conditions.

To functionally characterize the role of auxin homeostasis in general, and the specific roles of IAA biosynthesis gene *TAA1* and IAA degradation gene *DAO1*, the root hair responses to low P were characterized in loss of function *taa1* and *dao1* mutants. Our results revealed that, while there was a ~20 fold increase in root hair length under low P versus high P conditions in WT, this response was attenuated in *taa1* mutant roots by more than 60% (Fig. 2b, c). We also performed P experiments using a buffered nutrient system (see Methods section). Irrespective, we observed low P responsive root hair elongation in WT, whereas *aux1* and *taa1* almost completely disrupted this response (Supplementary Fig. 2). We also performed split P experiments where anchor roots (see Methods section) from the same plants were transferred to LP and HP media. In these experiments, WT roots on LP (but not HP) showed root hair elongation (Supplementary Fig. 3). In contrast, *taa1* roots grown on low P did not respond to low P conditions (Supplementary Fig. 3). These experiments revealed that root hair elongation under low P is a local adaptive response which is TAA1 dependent.

We also investigated the root hair phenotype of two different *DAO1* alleles recently described by Porco et al.[25]. The *dao1.2D* allele (in which the T-DNA is inserted 3′ of the coding sequence, increases *DAO1* expression)[25] had shorter hairs under P deficient conditions (Fig. 2d, e). Our observation reveals that root auxin levels limit root hair growth. In contrast, in the null *dao1-1* allele, root hairs were slightly longer compared to the wild type plants in both P sufficient and deficient conditions (Fig. 2d, e). Nevertheless, the absence of a strong phenotype in the loss of function allele *dao1.1* (Fig. 2d, e), questioned the importance of the auxin degradation pathway and/or may indicate redundancy with other hormone homeostasis mechanisms such as auxin conjugation, as reported by Porco et al.[25]. Taken together, our results suggest that

auxin synthesis, and possibly degradation, play an important role in promoting the local, adaptive root hair elongation response to low external P conditions.

**AUX1 mediates shootward auxin transport in low P conditions**. Auxin transport is also crucial for hair elongation in rice roots grown in low external P[23]. Given that mutations in the rice auxin influx transporter *OsAUX1* disrupt this root hair response under low P conditions[23], and Arabidopsis *aux1* mutants have been reported to have a root hair defect[26], we tested whether mutations in the orthologous *Arabidopsis AUX1* gene phenocopied this effect. Our results revealed that root hair elongation response under low P was severely disrupted in the loss of function *Arabidopsis* mutant *aux1-22*[27] compared to WT controls (Fig. 2b, c). Hence, AUX1-mediated auxin influx is required for the increased root hair elongation response to low external P in *Arabidopsis* as reported in rice[23].

Next, we identified the root tissue(s) in which AUX1-mediated auxin transport is required to facilitate the root hair elongation response to low external P. *AUX1* is predominantly expressed in several tissues at the apex of the *Arabidopsis* root, including proto-phloem, columella, lateral root cap (LRC), and epidermal cells[27–29]. *AUX1* expression in the LRC and epidermal cells has been reported to be important for root hair development under nutrient replete conditions[30]. We tested whether expressing the auxin influx carrier in a sub-set of its expression domain is sufficient to restore the *aux1* root hair elongation defect under low P conditions (Fig. 2b, c). We targeted expression of a HA-tagged *AUX1* sequence in *aux1-22* LRC and epidermal root cells using the tissue specific $_{aux1-22}$J0951»AUX1-HA line[31]. Expression in the LRC and epidermal cells was confirmed using an anti-HA antibody to reveal AUX1 spatial expression in root tissues after performing immuno-localization studies in the $_{aux1-22}$J0951»AUX1-HA line (Fig. 3a, b). Next, we tested whether J0951»AUX1-HA expression rescued the defective *aux1* root hair response under low P conditions. As shown in Fig. 3c, $_{aux1-22}$J0951»AUX1 rescued the *aux1* root hair elongation defect under low P conditions. Hence, *AUX1* expression in lateral root cap and epidermal cells is sufficient to rescue low P inducible root hair elongation. We reason that AUX1 is required to mobilize auxin from the root apex to hair differentiation zone (referred to as "shootward" auxin transport). Dindas et al.[24] have demonstrated that *AUX1* contributes more than 80% of the total root hair auxin uptake capacity. Hence, in the absence of auxin uptake carrier activity in *aux1*, the IAA-dependent root hair elongation response to low P conditions is severely perturbed.

**ARF19 regulates root hair elongation in low P conditions**. On reaching the root hair zone, auxin triggers a combination of short term and longer term signaling events. Dindas et al.[24] demonstrated that auxin triggered AUX1-dependent membrane depolarization. TIR1-mediated auxin perception and $Ca^{2+}$ signaling is closely coupled events in root hairs, occurring within seconds of each other. These rapid signaling events function to stabilize a calcium maxima close to the tip and promote integration of new cell wall materials. Longer term auxin responses like transcription are also required, to increase synthesis of new cell wall material, for example, to promote root hair elongation.

ARF7 and ARF19 are transcriptional activators[32] that regulate the majority of auxin responsive gene expression in seedling root tissues[33]. To investigate the role ARF7 and ARF19 during auxin-dependent root hair elongation responses to low P, we characterized the phenotypes of loss of function *arf7* and *arf19* mutant alleles. We observed that *arf7-1* seedlings exhibited a slight increase in root hair elongation compared to WT in low P

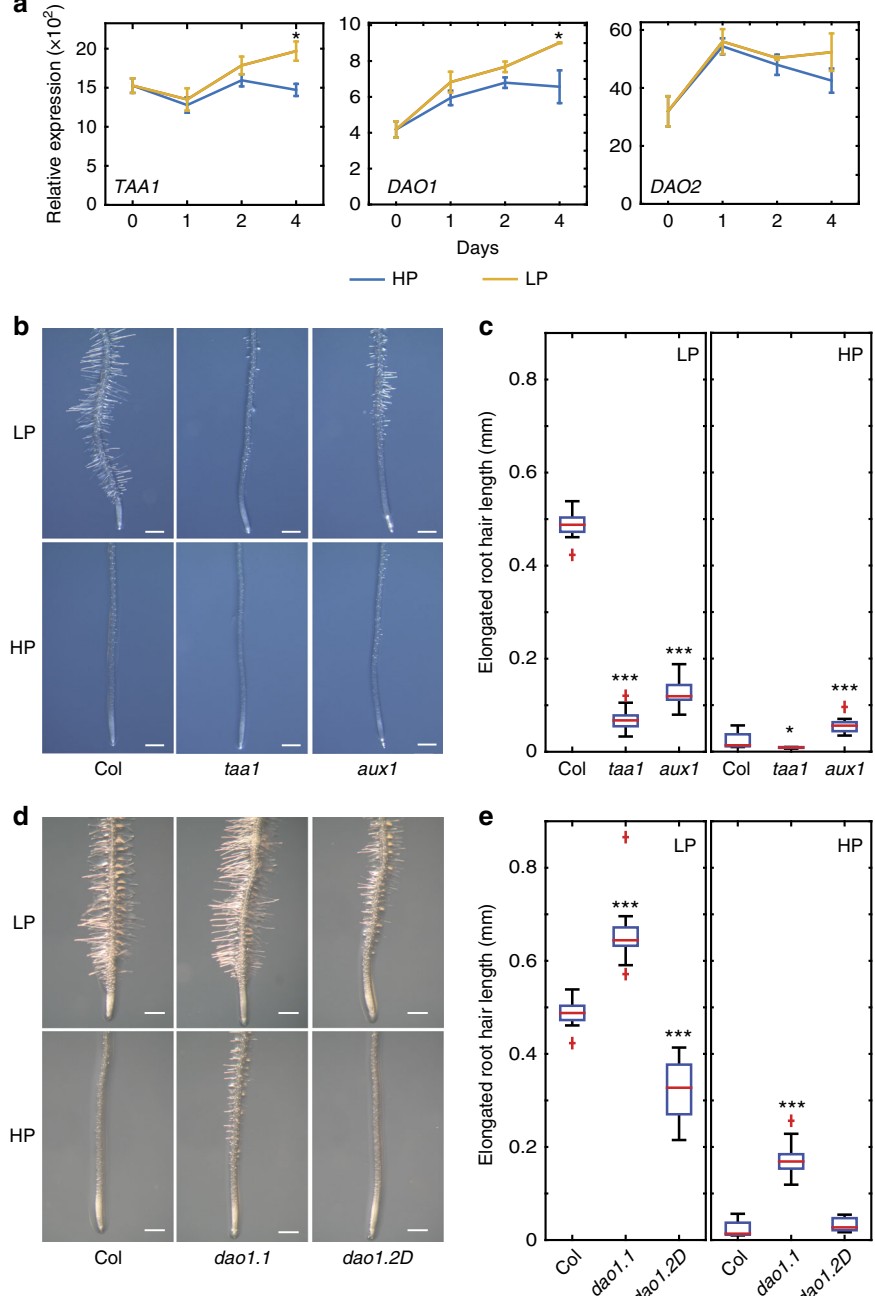

**Fig. 2** Auxin homeostasis is crucial for root hair elongation under low P. **a** Genes regulating Auxin biosynthesis (*TAA1*) and auxin degradation (*DAO1*, *DAO2*) are upregulated under low P (LP) conditions. *Indicate significant difference (*q*-value < 0.0001) judged by Student's *t*-test and after Benjamini and Hochberg false discovery rate correction. **b**, **c** Representative images (**b**) and boxplot (**c**) showing root hair length in the root hair zone under low (LP) or high P (HP) in Columbia (Col), *aux1 (aux1-22) and taa1* mutant seedlings. N = 3, scale bar = 0.5 mm. *, **, and *** indicate significant difference (*q*-value < 0.005, 0.0001 and 0.00001, respectively) judged by Student's *t*-test. **d**, **e** Representative images (**d**) and boxplot (**e**) showing root hair length under low (LP) or high (HP) P in Columbia (Col) and *dao1 (dao1.2 & dao1.2D)* mutant seedlings. N = 3, scale bar 0.5 mm. *, **, and *** indicate significant difference (*q*-value < 0.005, 0.0001, and 0.00001, respectively) judged by Student's *t*-test

conditions (Supplementary Fig. 4). In contrast, the promotion of root hair elongation under low P versus high P conditions was attenuated in *ARF19* knock out alleles (*arf19-a and arf19-b*; Fig. 4a–c) (Fig. 4b, c). Hence, our studies suggest that ARF19 (rather than ARF7) represents a key transcription factor promoting auxin dependent root hair elongation in response to low external P.

How is ARF19 activity enhanced under low P conditions? *ARF19* is reported to be strongly auxin-inducible[33]. Hence,

elevated auxin levels under low P conditions may increase *ARF19* transcript abundance. Consistent with this model, transcript profiling revealed that *ARF19* mRNA abundance increases under low P conditions (Fig. 4d). We validated this result using a reporter composed of the *ARF19* promoter driving the expression of 3 tandem copies of a nuclear localized VENUS fluorescent protein (*pARF19::3 × -Venus-NLS*). *pARF19*-driven reporter fluorescence was ~2 fold higher in low P versus high P conditions (Fig. 4e; Supplementary Fig. 5a). Expression of the *pARF19-*

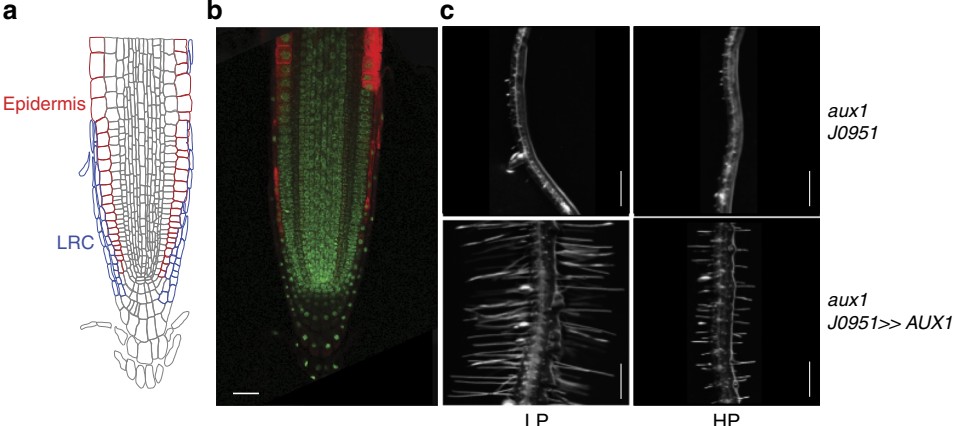

**Fig. 3** Expressing AUX1 in LRC and epidermal cells can restore *aux1* root hair defect under low P. **a** The schematic diagram of the *Arabidopsis* root apex. **b** Confocal image showing AUX1 expression in lateral root cap and epidermal cells in *aux1-22*J0951»AUX1 line. Scale bar = 20 μm. **c** Representative images showing root hair growth in the driver line *aux1-22*J0951 and *aux1-22*J0951»AUX1 lines under low (LP) or high P (HP). N = 3, scale bar = 0.5 mm

driven reporter also extended throughout the root hair differentiation zone under low P conditions. We conclude from our mutant, transcriptomic, and reporter studies that ARF19 represents a key regulator for auxin-inducible root hair elongation under low P conditions.

**Transcription factors RSL2 and RSL4 are induced in low P.** ARF19 is likely to promote root hair elongation by regulating the expression of genes like *RSL2* and *RSL4* that encode basic Helix-Loop-Helix transcription factors that promote root hair initiation and elongation[19]. We initially tested whether *rsl4* mutant lines disrupt the low P root hair elongation response. Root hair lengths in *rsl4-1* were shorter than WT (Fig. 5a, b). This mutant phenotype suggests that RSL4 functions as part of the low P inducible root hair elongation machinery. However, the low P root hair defect exhibited by the *rsl4-1* allele was not as severe as observed with the *arf19* mutant alleles (Fig. 4b, c). This implies that ARF19 regulates other genes, in addition to *RSL4*, to promote root hair elongation under low P conditions. Indeed, transcript profiling revealed that mRNAs for other closely related members of the bHLH family (such as *RSL2*) are up-regulated under low P conditions (Fig. 5c; Supplementary Fig. 6a). To test the role of RSL2 in regulating root hair elongation in low P, we characterized its loss of function mutant phenotype. This revealed a striking defect in the *rsl2* mutant root hair response to low P, which was even more severe than *rsl4* (Fig. 5c). Next, we monitored its expression using an *RSL2:GFP* reporter line. This also revealed an increase in the RSL2:GFP signal under low P conditions.

*RSL4* has been shown to be induced by auxin[19]. We searched for the presence of auxin response element(s) in the *RSL4* (and *RSL2*) promoter sequence. Motif analysis revealed highly conserved auxin response elements in both *RSL* promoters (Supplementary Fig. 6b) that represent target(s) for ARFs such as ARF19. Consistent with the *bHLH* gene being a target for regulation by ARF19, we observed the sequential induction of *ARF19* and then *RSL4* mRNAs in our transcriptomic time course dataset (Fig. 4d and Supplementary Fig. 5b). RSL4 protein abundance is correlated with root hair elongation[20]. This prompted us to monitor RSL2 and RSL4 levels under low P versus high conditions employing GFP fusion proteins driven by their native promoters. RSL2 and RSL4 levels are higher (Supplementary Fig. 5b) under low P than high P conditions (Fig. 5d and ref.[19]). Spatially, the RSL2 and RSL4 GFP reporters overlap with the *pARF19*-driven GFP reporter in the root hair

differentiation zone (Fig. 5c), consistent with these genes being targets for regulation by ARF19 under low P conditions.

## Discussion

Developmental plasticity is crucial for plant roots to optimize foraging for resources in highly heterogeneous soil environments[34]. Root hair elongation represents an important adaptive response to acquire immobile nutrients such as P[6,23]. We report that *Arabidopsis* roots exposed to low external P conditions employ the hormone auxin to enhance uptake of this key nutrient by promoting root hair elongation.

Our study reveals that the promotion of root hair elongation under low external P requires the concerted activity of auxin synthesis, transport and transcription-related components, from which we have been able to construct a mechanistic framework for this important root adaptive response pathway. Using the DR5 based auxin response reporter and direct auxin measurements we show that IAA accumulates in the root hair zone when grown in low P conditions (Fig. 1b, c). Currently it is not clear where IAA is synthesized in response to low P conditions. Transcriptomic and genetic approaches reveal that TAA1-is required for this low P induced root hair response. *TAA1* encodes a key enzyme in auxin biosynthesis[35], which is expressed in the root apex and induced under low P conditions (Fig. 2a). The *taa1* mutant disrupts root hair elongation under low P (Fig. 2b, c). Taken together these results suggest that TAA facilitates auxin synthesis under low P conditions presumably in the root apex. Split root experiments suggest that root hair elongation is a local response and is independent of plant P status[36,37]. As *TAA1* is expressed in the root apex and IAA can be synthesized in the root apex[35,38], it is likely that there is local auxin synthesis in the root apex under low P. Consistent with this conclusion, targeted expression studies in *Arabidopsis* revealed that low P induced hair elongation required IAA to be mobilized via the AUX1 influx carrier from the root apex to the root differentiation zone (Fig. 3). In a co-submitted paper, Giri et al.[23] reported that this was also the case in rice roots. Our observations made in *Arabidopsis* and rice reveal the functional conservation of the AUX1-dependent shootward auxin transport pathway (from the root apex to hair differentiation zones) for this important root P adaptive response in these distantly related angiosperm species.

On reaching the root hair zone, AUX1-dependent IAA uptake triggers very rapid (i.e., seconds to minutes) and longer term (i.e., minutes to hours) signaling processes. Dindas et al.[24] reported that the rapid auxin-dependent root hair response to low P relied

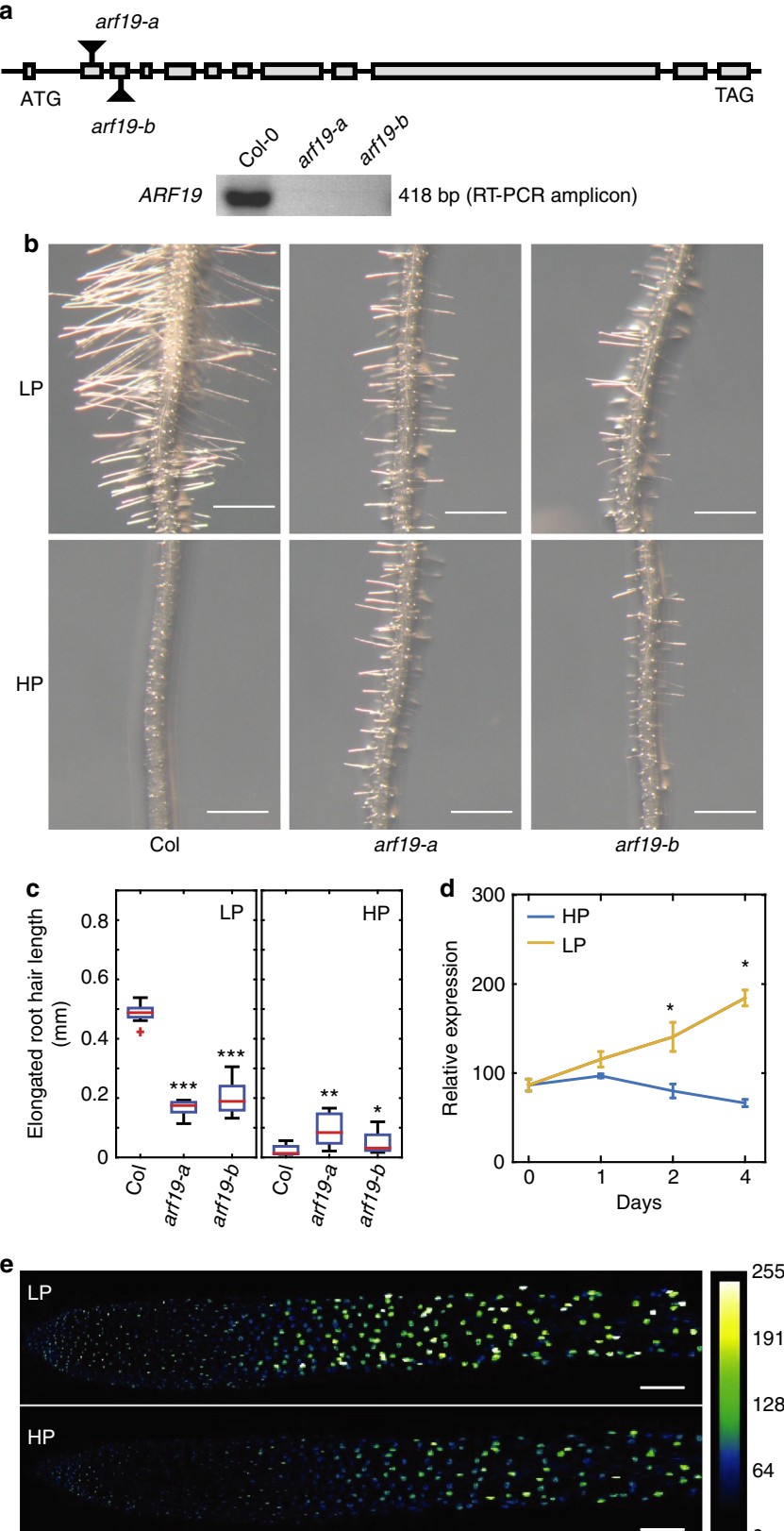

**Fig. 4** Auxin Response Factor *ARF19* regulates root hair elongation under low P in *Arabidopsis*. **a** Cartoon showing positions of two different *arf19* mutant alleles used in this study (top) and Reverse Transcription-PCR results showing that *arf19-a* and *arf19-b* are null alleles. **b** Representative images showing root hair growth of Columbia (Col), *arf19-a* and *arf19-b* seedlings under low P (LP) or high P (HP) conditions. $N = 3$, scale bar = 0.5 mm. **c** Boxplot showing root hair growth of Columbia (Col), *arf19-a* and *arf19-b* seedlings under low P (LP) or high P (HP) conditions. *, **, and *** indicate significant difference ($q$-value < 0.005, 0.0001, and 0.00001, respectively) judged by Student's $t$-test. **d** Expression profiling studies showing *ARF19* expression under low P (LP) or high P (HP) conditions. *Indicates significant difference ($q$-value < 0.05) judged by Student's $t$-test and after Benjamini and Hochberg false discovery rate correction. **e** Fluorescence intensity of *pARF19::3XVENUS-NLS* seedlings grown under low P (LP) and high P (HP). $N = 3$, scale bar = 100 μm

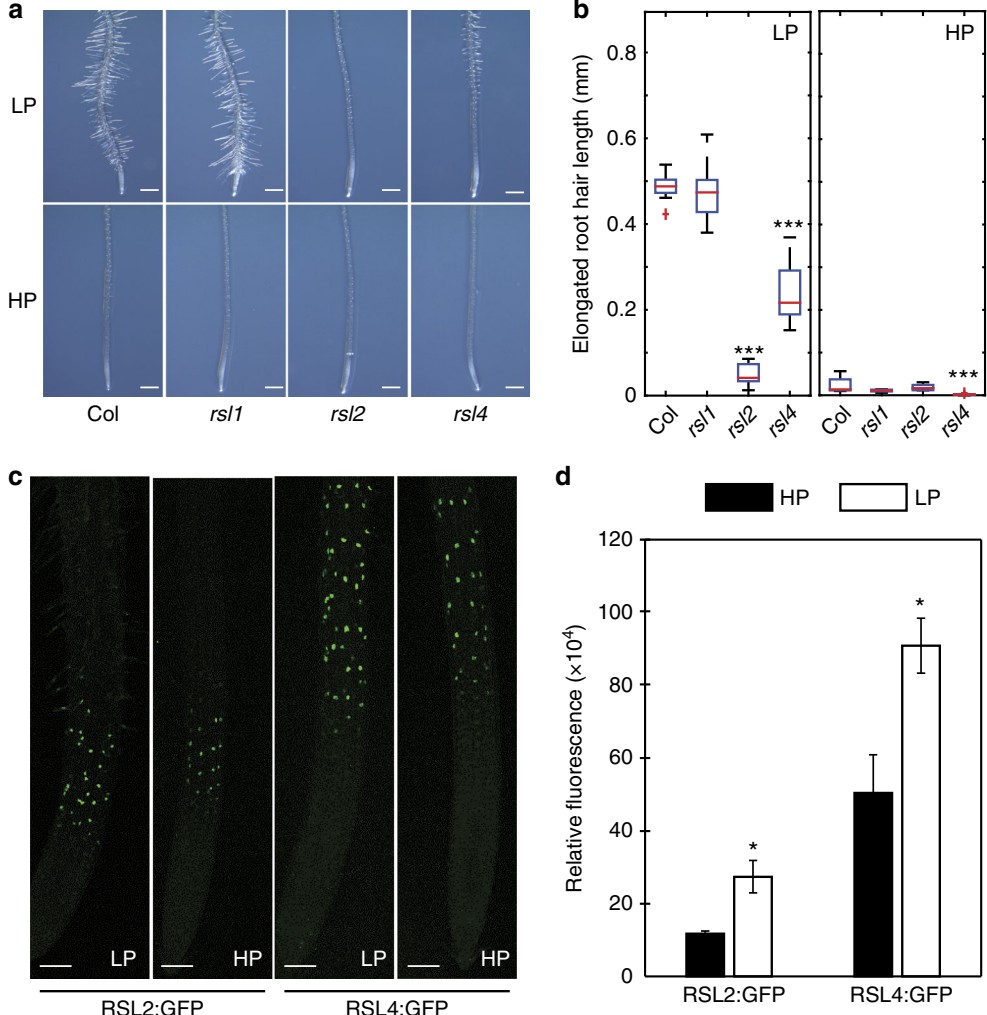

**Fig. 5** bHLH transcription factors *RSL2* and *RSL4* are involved in low P responsive RH elongation in *Arabidopsis*. **a, b** Representative images (**a**) and boxplot (**b**) showing root hair growth of Columbia (Col), *rsl1*, *rsl2*, and *rsl4* mutant seedlings under low P (LP) or high P (HP) conditions. *, **, and *** indicate significant difference *q*-value < 0.005, 0.0001, and 0.00001, respectively, judged by Student's *t*-test. N = 3, scale bar = 0.5 mm. **c**. Fluorescence intensity of pRSL2::GFP-RSL2 and pRSL4::GFP-RSL4 seedlings grown under low P (LP) and high P (HP). N = 3, scale bar = 100 μm. **d** Measured raw integrated fluorescence intensities of pRSL2::GFP-RSL2 and pRSL4::GFP-RSL4 grown for 5 days under low (LP, empty bars) and high (HP, black bars) P conditions. *Indicates significant difference (*q* value < 0.05), judged by Student's *t*-test

on TIR1 to promote hair elongation via elevating intracellular auxin and calcium signaling. Maintenance of calcium maxima at the root tip is known to be essential for hair growth to coordinate secretion, endocytosis, and actin dynamics[39]. Fascinatingly, Dindas et al.[24] observed that the calcium maxima at the root hair tip was transient and dependent on auxin. Hence, to maintain this tip focused calcium gradient and promote hair elongation would require a sufficient duration and level of auxin, as experienced during root exposure to low external P. Increased auxin in root hair cells will also trigger longer term transcriptional responses. This increase in auxin response could help explain why the root hair defects in the auxin mutants *axr1* and *axr2* under control conditions, can be rescued by low external P treatment[40]. The auxin-inducible transcription factor ARF19 appears to play a critical role regulating the expression of key genes that promote root hair elongation under low P conditions, since this response is abolished in *arf19* mutant alleles (Fig. 4c). Yi et al.[19] reported that auxin acts primarily via the transcription factor RSL4 to promote root hair elongation. A number of downstream targets have recently been identified[16,41]. This includes Ca-dependent Protein Kinase 11 (CPK11), which could

function in a calcium-dependent signaling pathway mediating root hair elongation. Nevertheless, only a third of the 90 root hair genes that were found to be auxin responsive were affected in *rsl4* mutants[19,41,42]. Moreover, the existence of RSL4-dependent and independent auxin mediated root hair response pathways is consistent with the reduced severity of the *rsl4-1* defect in low P compared to *rsl2* and *arf19* alleles (Figs. 4c and 5b).

In summary, we have constructed a mechanistic framework for the low P induced root hair elongation response pathway. We demonstrate that low P elevates IAA levels in the root apex facilitated by TAA1 mediated auxin synthesis and AUX1 dependent auxin transport. The resulting increase in IAA levels leads to the induction of *ARF19* in the root apex resulting in *RSL2* and *RSL4* induction in the the elongation and differentiation zones respectively promoting root hair elongation (Fig. 6).

## Methods

**Plant materials**. *Arabidopsis* ecotype Columbia was used as the wildtype in all experiments. *arf19* mutant alleles *arf19-a* (SAIL_92_G09, N872822) and *arf19-b* (SK-24322, N1007173) were obtained from the Nottingham Arabidopsis Stock Centre and the insertion was confirmed by PCR using forward and reverse primers (obtained using T-DNA primer design tool) in combination with LB3 (for arf19-a)

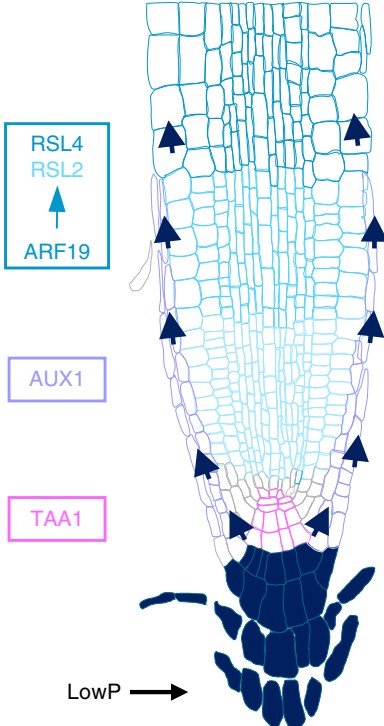

**Fig. 6** Model for root hair elongation under low P. Low P elevates IAA levels in the root apex facilitated by TAA1 mediated auxin synthesis and AUX1 dependent auxin transport. The resulting increase in IAA levels leads to the induction of ARF19 in the root apex resulting in RSL2 and RSL4 induction in the elongation and differentiation zones, respectively, promoting root hair elongation

and pSKTAIL-L1 (for arf19-b) border primers (Supplementary Table 1). *ARF19* expression in these alleles was checked by Reverse Transcription-PCR using primers ARF19_RT_For and ARF19_RT_Rev (Fig. 4a and Supplementary Table 1). Following mutant alleles: *arf7-1*, *arf19* (*arf19-a* and *arf19-b*), *rsl2*, *rsl4-1* (back-crossed four times), *aux1* (aux1-22), *taa1*, *dao1.1*, and *dao1.2*D and reporters: *pARF19::3XVENUS-NLS*; *pRSL2::GFP-RSL2*, and *pRSL4::GFP-RSL4* were used in phosphate treatment experiments.

**Growth conditions**. Seeds were surface sterilized with 70% ethanol for 10 min and 100% ethanol for 30 s followed by five washes with sterile water and stratified at 4 °C for 48 h in dark. For phosphate treatment experiments, seeds were germinated for 5 days on vertical 1/4th MS (Murashige and Skoog) Plates (supplemented with 1% agar, at pH 5.7) in growth room (22 °C; continuous light; 100–125 μmol/m$^2$/s). 5-day-old seedlings were transferred to MS salt (without phosphate) plates supplemented with 1% agar and 3 μM (low) and 312 μM (high) P. pH of media was set to 5.7 and low P media was supplemented with equimolar concentration of KCl. We also performed P experiments using a buffered nutrient system. Plants were grown under sufficient (MB, 50 μM soluble P) or limiting (LB, 3 μM soluble P) conditions with a buffered phosphorus delivery regime. Plants were grown for 3 days (16 h day, 100–120 μmol min$^{-2}$) on 1/4 strength MS media without sucrose before being transferred to the respective treatment. After four days of growth, images were obtained on a dissecting scope and root hair lengths were measured. Ten plants per genotype, per treatment were measured. Error bars represent the standard error of the mean.

**Transcriptome studies**. For microarray experiments, plants were grown on 0.5 MS Difco agar as described previously using 625 and 100 μM phosphate for control and P-deficient conditions, respectively. RNA was extracted from root material using an RNeasy plant mini kit (Qiagen) with on-column DNase treatment according to the manufacturer's instructions. cDNA was prepared using random hexamer primers and SuperScript II reverse transcriptase (Life Technologies) according to the manufacturer's protocol. RNA amplification, labeling and hybridization to Agilent microarrays (*Arabidopsis* (V4, 021169) Gene Expression Microarray) were conducted following the manufacturer's protocol (Agilent Technologies). Analysis of microarray data was performed with the R package limma[43]. Raw feature intensities were background corrected using "normexp," and "quantile" normalization method was used to normalize between arrays.

Differential expression was performed by fitting a linear model to log2-transformed data by an empirical Bayes method[44]. The Benjamini and Hochberg method was used to correct for multiple testing. The transcriptome raw data is deposited in electronic data archive library (e!DAL—https://edal.ipk-gatersleben.de) repository under the accession code https://doi.org/10.5447/IPK/2018/4.

**Auxin estimation**. Roots tip (~1 mm) and root hair zone (next 1.5 mm region) from 9-days-old seedlings grown on low and high P media were cut using a sharp scissor and snap frozen in liquid nitrogen. Seventyfive roots were collected per replicate with four replicates per experiment. Five-hundred picograms of $^{13}C_6$-IAA internal standard was added to each sample before purification. Auxin estimation was done as described previously[45], with minor modifications.

**GUS quantification**. GUS intensity was measured using imageJ software. Briefly, Leica image in imageJ/Fiji software was imported and "color tool" in image tab was used to split channels and convert single image into three grayscale images. Red channel image was selected and "adjust tool" in image tab was used to assign values to pixels between 0 and 255. Intensity threshold was used to adjust the pixels in the selected range (corresponds to GUS staining) that are highlighted in red. Raw integrated density (i.e., the sum of the values of the pixels in the selection) was measured using the usual measurement option (Analyze/Measure menu option) with measurements limited to only the selected pixels.

**DNA constructs and transgenic materials**. *pRSL2::GFP-RSL2*[19] and *pRSL4::GFP-RSL4*[19] transgenic lines were obtained from Liam Dolan, Department of Plant Sciences, University of Oxford, Oxford, OX1 3RB, UK.
The *pARF19-3xmVenus-N7ARF19* was cloned using Gateway Multisite cloning technology. The ARF19 fragment (−4906 to +452 bp) was PCR amplified using primers ARF19_For and ARF19_Rev (Supplementary Table 1) and cloned into pDONR P4-P1R to create pARF19 pDONR P4-P1R. The later was recombined in a multisite Gateway reaction with the following plasmids: 3 × mVenus-N7 pDONR211 (containing triple mVenus coding sequences and N7 nuclear localization signal), OCS terminator pDONR P2R-P3 (containing the stop codon followed by octopine synthase (OCS) terminator), and pK734GW (the destination vector containing selection kanamycin resistance gene for in planta selection). The resulting final plasmid was transformed into *Agrobacterium tumefaciens* C58pMP90 strain by electroporation and then transformed into Col-0 plants by floral dip method[46].

**Data availability**. The transcriptome raw data is deposited in electronic data archive library (e!DAL—https://edal.ipk-gatersleben.de) repository under the accession code https://doi.org/10.5447/IPK/2018/4. Other data and materials supporting the findings of this study may be obtained from the corresponding authors upon reasonable request.

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

## Acknowledgements

This work was supported by the awards from the Biotechnology and Biological Sciences Research Council [grant numbers BB/G023972/1, BB/R013748/1, BB/L026848/1, BB/M018431/1, BB/PO16855/1, BB/M001806/1, BB/P010520/1]; the European Research Council FUTUREROOTS Advanced Investigator grant [grant number 294729]; Leverhulme Trust [grant number RPG-2016-409]; Royal Society [grant number WM130021, NA140281]; Newton International Fellowship (NF140287) and British Council Newton Bhabha (228144076). This work was also supported by funds from the University of Nottingham Future Food Beacon of Excellence Nottingham Research and PhD+ fellowship schemes; the Interuniversity Attraction Poles Program initiated by the Belgian Science Policy Office [P7/29]; the Swedish Governmental Agency for Innovation Systems (VINNOVA), and the Swedish Research Council (V.R.). NSF-MCB1158181 and a joint studentship between the University of Nottingham and Institut National de la Recherche Agronomique (INRA). We also thank Roger Granbom (Swedish University of Agricultural Sciences) for skilful technical assistance.

## Author contributions

R.B., J.G, B.K.P., R.F.H.G., A.H., R.T., J.T., N.L., M.H., K.S., A.R., U.V., J.A., A.S., and J.Y. performed the experiments and contributed the experimental data; R.B., J.G., B.K.P., K.L., K.M.B., J.P.L., L.D., T.V., A.B., D.W., N.W., R.S., and M.J.B. designed experiments; and R.B., J.G., B.K.P., R.S., and M.J.B. wrote the manuscript.

## Additional information

**Competing interests:** The authors declare no competing interests.

