## [Peer Review File · Nature Communications]

Reviewers' comments:

Reviewer #1 (Remarks to the Author):

This manuscript addresses a biologically interesting and agronomical important topic, how phosphate availability modulates root hair elongation, which is an important trait in determining the capacity of the plant to explore the soil and scavenge phosphate. Although the work is important and the experiments well executed, the paper is done with a suit of previously describe mutants and reporter lines and mainly report on the independent discovery of several auxin related processes that were previously described by other research groups. I will try to explain this point in the following paragraphs:

1 The effect of root hair elongation in the results section is referred to figure 1A for Arabidopsis and to the co-submitted paper of Giri et al. The effect of low Pi availability on root hair elongation has been previously described in several papers by the group of one of the authors, Jonathan Lynch, and other research groups, papers that must be cited. The effect of low Pi on root hair elongation has also been previously reported in several papers, including from one from the late Ping Wu's group (figure 7; Plant Phys 1467: 1673-1686). Thus, neither in Arabidopsis nor rice is the effect of low Pi on root hair elongation is not a novel finding and proper references must be cited.

2 The role of auxin on root hair elongation is well documented since over 20 years and an increase transport and accumulation of auxins in the root tip of Pi-starved seedlings was previously reported (Plant Phys 138: 2061-2074). Although the quantification of auxin in the root tip and root hair elongation zone is a nice additional data reported in this paper. Mass spec imaging analysis of the distribution of auxin in the root tip in both Pi sufficient and deprived plant would be needed to more precisely determine in which cells auxin accumulate.

3 Since it has been well documented that auxin plays a fundamental role in root hair formation and elongation, it is hardly surprising to find that root hair elongation is partially impaired in response to low Pi in the auxin biosynthesis *taa1* and auxin transport *aux1*. It was previously reported that *aux1* and *tta1* mutants have short root hairs and that both mutants increase root hair when treated by auxin. Moreover, since *taa1* and *aux1* are affected in both low and high Pi and both mutants still are responsive to low Pi in terms of root hair elongation, their thus phenotype seems to be independent of the low Pi-response.

4 As presented and this paper and previously reported in other global transcriptional studies of the Arabidopsis, TAA1 is induced in response to low phosphate leading to an increase production of auxins, which in turn result in longer root hairs in Pi-deprived seedlings. Thus, it is not surprising that mutation in auxin biosynthesis affect root hair elongation independently of the plant nutritional status. An important finding it would be that TAA1 specifically produces auxins in certain cells to be transported to promote root hair elongation, but that would require mass spectrometry imaging analysis to determine in which cells, the *taa1* mutants affects auxin concentration, because if TAA1 is involved in general auxin synthesis an unspecific effect is expected in root hair elongation in response to low Pi.

5 The role of AUX1 on root hair elongation was previously reported. It was shown that *aux1* seedlings have short root hair which increase size upon treatment with external auxins (Plant Journal 16: 553-560). Thus, it is logic that in *aux1* root hair elongation in response to low Pi is also indirectly affected. This is supported by the finding that *aux1* seedlings have shorter hairs that the WT in both high and low Pi, but still responsive to low Pi availability.

6 An interesting piece of data is presented in figure 3, in which it is shown that expression of AUX1 under control of the Jo951 trans activator fully complements root hair formation in both low and high Pi. These data show that expression of AUX in the lateral root cap and epidermal cells is sufficient to promote root hair elongation. However, since constitutive expression of AUX1 in root cap and epidermal cells leads to a fully functional response to low Pi, these data shows elegantly shows the role of AUX1 in root hair elongation but not a role in regulating the response to low Pi, as stated in the

title of this paper "A mechanistic framework for auxin dependent root hair elongation in response to low external phosphate"

7 At the end of the results section the authors report the effect of mutations in ARF19 and ARF7 on the root hair response to low pi. Again, it is not surprising that mutations in ARF19 have an impact on root hair elongation in response to low Pi given that ARF19 affects most if not all the root auxin responses, including lateral root formation and elongation.

In summary, I find that the work reported in this manuscript is interesting and the experiments well designed and carried out, but that the main findings are of a confirmatory nature of the role of the auxin synthesis, transport and signaling genes in root hair elongation, which was previously been reported in several papers describing root hair formation and elongation in both Arabidopsis and rice. The root hair response to low Pi is a local response that is independent from the processes regulated by systemic master regulators PHR1/SPX, therefore, identifying the components that regulate this local response would be a major advance in the understanding of the mechanisms that regulate root hair elongation in response to low Pi. Given the confirmatory nature of the data presented and the lack of novel insights on the mechanisms that regulate the expression of TAA1, AUX1 and other auxin response components, I find that this paper lacks essential components to be published in a high impact journal such as Nature Communications.

Reviewer #2 (Remarks to the Author):

A mechanistic framework for auxin dependent Arabidopsis root hair elongation in response to low external phosphate by Rahul Bhosale et al. is one of three manuscripts that have been co-submitted. I am really enthusiastic about the message of the stories together; they each show a different angle of a multi-faceted message that links root positioning and root hair growth with auxin synthesis and directed transport. The studies claim to have identified a major regulatory mechanism that controls traits of agronomic importance. Each of the stories have their own focus and interesting points. It describes a series of elegant experiments that support a clever line of thought.

This manuscript describes an AUX1 based mechanism for regulation of root hair length in response to phosphate concentration.

I have serious issues with the interpretation of the data presented, mainly those in Fig. 2. Firstly, I find it hard to believe that the root hair lengths given in panel C do not differ significantly between the wild type, aux1-22 and taa1 under both LP and HP and in panel E do not differ between wild type and dao1.1 under both LP and HP. The differences in root hair length seem enormous, but the lack of any information makes it impossible to interpret the data. Do the bars represent SE or SD? How many root hairs were measured? How many biological replicates? What statistical tests were performed? What were the p values of the insignificant differences?

Secondly, although the authors claim that there are no significant differences in root hair length under HP conditions, a quick measurement of the bar lengths in panel C reveals that the root hair length increase when grown on LP is 3-fold for both wild type and taa1, and 2.5-fold for aux1-22. As I argued in my review of the manuscript by Giri et al., root hairs of all 3 genotypes respond to LP by increasing their root hair length ~3-fold. Thus, the scale of the root hair response to LP is similar in all 3 genotypes, suggesting that the mutated genes have no effect on P perception. One could use a similar line of reasoning for the dao1.1 mutant: the authors are correct in claiming that the 10-fold increase in root hair length of wild type plants is reduced in dao1.1 plants. However, the root hairs of dao1.1 are so much longer than wild type root hairs under HP conditions, that, to increase 10-fold in length, they should reach lengths that are at least 2-fold longer than the maximum length that has been

observed in wild type root hairs under any condition. This suggests that genetic factors controlling maximal root hair length, and not a decrease in *dao1* activity, are responsible for the reduced increase in root hair length.

In Fig 1e., part of the max. intensity projection appears to be missing. There appears to be selective DR5:NLS-Venus expression in median epidermal/cortical cell files, while expression is absent in the upper, more central cell files. In addition, the microscopy data would benefit from a quantitative analysis. I could argue that a slight difference in focal plane or root curvature between C and D is causing the decreased fluorescence in the centre of the roots. I would recommend the authors to quantify their data, provide the number of replicates and use internal controls (for example propidium iodide staining intensities) to correct for, for example, differences in working distance.

The data in Fig. 4 and Fig. 5 should be quantified, but appears convincing and promising.

Reviewer #1 (Remarks to the Author):

This manuscript addresses a biologically interesting and agronomical important topic, how phosphate availability modulates root hair elongation, which is an important trait in determining the capacity of the plant to explore the soil and scavenge phosphate. Although the work is important and the experiments well executed, the paper is done with a suit of previously describe mutants and reporter lines and mainly report on the independent discovery of several auxin related processes that were previously described by other research groups. I will try to explain this point in the following paragraphs:

1 The effect of root hair elongation in the results section is referred to figure 1A for Arabidopsis and to the co-submitted paper of Giri et al. The effect of low Pi availability on root hair elongation has been previously described in several papers by the group of one of the authors, Jonathan Lynch, and other research groups, papers that must be cited. The effect of low Pi on root hair elongation has also been previously reported in several papers, including from one from the late Ping Wu's group (figure 7; Plant Phys1467: 1673-1686). Thus, neither in Arabidopsis nor rice is the effect of low Pi on root hair elongation is not a novel finding and proper references must be cited.

To clarify, we are certainly not trying to claim root hair elongation under low P is a novel finding of this manuscript. Indeed, we have cited several references showing a connection between low P and root system architecture (references 6-13) and have also cited work by one of our co-authors Jonathan Lynch showing a link between root hair elongation and low P availability (reference 6 (and 4)). We are also familiar with Ping Wu's work and have now cited this paper in the revised manuscript.

2 The role of auxin on root hair elongation is well documented since over 20 years and an increase transport and accumulation of auxins in the root tip of Pi-starved seedlings was previously reported (Plant Phys 138: 2061-2074). Although the quantification of auxin in the root tip and root hair elongation zone is a nice additional data reported in this paper. Mass spec imaging analysis of the distribution of auxin in the root tip in both Pi sufficient and deprived plant would be needed to more precisely determine in which cells auxin accumulate.

We completely agree that the link between auxin and root hair elongation has been made by several groups over the past 20 years, including our own with Claire Grierson. However, the novel finding we report in our manuscript is the role of auxin promoting root hair elongation in response to low external P levels. Whilst the paper by Nacry et al (in Plant Phys 138: 2061-2074) reported how low P causes changes in auxin response at the root apex that underpin root systems architecture changes, it focused on modifications to primary and lateral root growth and development, rather than root hair elongation. In addition, the molecular basis for the increase in auxin response was also not determined in Nacry et al. We have exploited the rich source of auxin biosynthesis and degradation genes identified since the Nacry paper, to reveal the important role of TAA1 in increasing auxin abundance at the root apex. We also reported in Fig 1B using mass spec analysis to directly measure auxin accumulation in the root tip and root hair zones under low P. In parallel, figure 1 (panels C-H) shows how we used auxin response reporter DR5:VENUS to provide a cellular level of resolution of auxin accumulation under low P.

3 Since it has been well documented that auxin plays a fundamental role in root hair formation and elongation, it is hardly surprising to find that root hair elongation is partially impaired in response to low Pi in the auxin biosynthesis *taa1* and auxin transport *aux1*. It was previously reported that *aux1* and *tta1* mutants have short root hairs and that both mutants increase root hair when treated by auxin. Moreover, since *taa1* and *aux1* are affected in both low and high Pi and both mutants still are responsive to low Pi in terms of root hair elongation, their thus phenotype seems to be independent of the low Pi-response.

As the reviewer themselves notes, no previous paper had previously made the link between auxin and root hair elongation in response to low P. Hence, our manuscript reports the novel finding that auxin is required for root hair elongation under low P. We also provide a comprehensive molecular and cellular framework in which components mediating auxin synthesis, transport and response pathways are positioned that mediate this adaptive response. Please note that *taa1* and *aux1* mutations result in major disruptions to the low P induced RH elongation response compared to WT (as seen in Fig 2B & C), consistent with playing an important role during this low P adaptive root response.

4 As presented in this paper and previously reported in other global transcriptional studies of the Arabidopsis, TAA1 is induced in response to low phosphate leading to an increase production of auxins, which in turn result in longer root hairs in Pi-deprived seedlings. Thus, it is not surprising that mutation in auxin biosynthesis affect root hair elongation independently of the plant nutritional status. An important finding it would be that TAA1 specifically produces auxins in certain cells to be transported to promote root hair elongation, but that would require mass spectrometry imaging analysis to determine in which cells, the *taa1* mutants affects auxin concentration, because if TAA1 is involved in general auxin synthesis an unspecific effect is expected in root hair elongation to low Pi.

We have demonstrated using a split root system that TAA1 function is required for roots to respond locally (rather than systemically) to low external P and form elongated root hairs since (unlike WT) the *taa1* mutant failed to undergo RH elongation when its roots were exposed to low P (Fig. S3). In parallel, we have employed a genomic TAA1-GUS reporter to reveal that the protein's abundance increases in root apical tissues in response to low P (Fig. S1), including under split root conditions. When combined with our original mass spec measurements, our new genetic and reporter data provides evidence that low P conditions triggers TAA1 mediated changes in its level that alters auxin levels and results in changes in root hair elongation.

5 The role of AUX1 on root hair elongation was previously reported. It was shown that *aux1* seedlings have short root hair which increase size upon treatment with external auxins (Plant Journal 16: 553-560). Thus, it is logic that in *aux1* root hair elongation in response to low Pi is also indirectly affected. This is supported by the finding that *aux1* seedlings have shorter hairs than the WT in both high and low Pi, but still responsive to low Pi availability.

The reviewer is correct that several papers have reported AUX1 regulated root hair elongation including the original Plant J paper by Pitts et al (1998) and our later Nature Cell Biology paper with Claire Grierson (Jones et al, 2008). However, as discussed in response to point 3 above, none of the earlier papers have made the connection between AUX1, auxin and low P in this fascinating root hair local adaptive response and which is one of the novel findings of our current study.

6 An interesting piece of data is presented in figure 3, in which it is shown that expression of AUX1 under control of the Jo951 trans activator fully complements root hair formation in both low and high Pi. These data show that expression of AUX in the lateral root cap and epidermal cells is sufficient to promote root hair elongation. However, since constitutive expression of AUX1 in root cap and epidermal cells leads to a fully functional response to low Pi, these data shows elegantly shows the role of AUX1 in root hair elongation but not a role in regulating the response to low Pi, as stated in the title of this paper "A mechanistic framework for auxin dependent root hair elongation in response to low external phosphate"

We demonstrate in Fig. 3 how AUX1 plays a key role in Arabidopsis roots (as described in rice in Giri et al, co-submitted) transporting auxin from the apex to hair elongation zone. This is entirely consistent with the site of TAA1 up-regulation by low P being in the root apex, necessitating this hormone to be mobilised to the root hair zone.

7 At the end of the results section the authors report the effect of mutations in ARF19 and ARF7 on the root hair response to low pi. Again, it is not surprising that mutations in ARF19 have an impact on root hair elongation in response to low Pi given that ARF19 affects most if not all the root auxin responses, including lateral root formation and elongation.

Unlike *arf7* or *arf7 arf19* mutants, very few reports have described a root phenotype for the single *arf19* mutant. Hence, we were surprised to discover that *arf19* had a much stronger root hair phenotype than *arf7*, particularly since ARF7 regulates ARF19 expression.

In summary, I find that the work reported in this manuscript is interesting and the experiments well designed and carried out, but that the main finding are of a confirmatory nature of the role of the auxin synthesis, transport and signaling genes in root hair elongation, which was previously been reported in several papers describing root hair formation and elongation in both Arabidopsis and rice. The root hair response to low Pi is a local response that is independent from the processes regulated by systemic master regulators PHR1/SPX, therefore, identifying the components that regulate this local response would be a major advance in the understanding of the mechanisms that regulate root hair elongation in response to low Pi. Given the confirmatory nature of the data presented and the lack of novel insights on the mechanisms that regulate the expression of TAA1, AUX1 and other auxin response

components, I find that this paper lack essential components to be published in a high impact journal such as Nature Communications.

We would argue in the strongest possible terms that our original manuscript provided a novel mechanistic framework explaining how auxin and its synthesis, transport and response components regulate root hair elongation in response to low P. Moreover, our new split root experiments provide compelling evidence that auxin-related components like TAA1 regulate this local P response, which the reviewer acknowledges “*would be a major advance in the understanding of the mechanisms that regulate root hair elongation in response to low Pi.*” Hence, we would reason that the revised manuscript merits re-consideration for publication by *Nature Commnication*.

Reviewer #2 (Remarks to the Author):

A mechanistic framework for auxin dependent Arabidopsis root hair elongation in response to low externa phosphate by Rahul Bhosale et al. is one of three manuscripts that have been co-submitted. I am really enthusiastic about the message of the stories together; they each show a different angle of a multi-faceted message that links root positioning and root hair growth with auxin synthesis and directed transport. The studies claim to have identified a major regulatory mechanism that controls traits of agronomic importance. Each of the stories have their own focus and interesting points. It describes a series of elegant experiments that support a clever line of thought.

This manuscript describes an AUX1 based mechanism for regulation of root hair length in response to phosphate concentration.

I have serious issues with the interpretation of the data presented, mainly those in Fig. 2. Firstly, I find it hard to believe that the root hair lengths given in panel C do not differ significantly between the wild type, *aux1-22* and *taa1* under both LP and HP and in panel E do not differ between wild type and *dao1.1* under both LP and HP. The differences in root hair length seem enormous, but the lack of any information makes it impossible to interpret the data. Do the bars represent SE or SD? How many root hairs were measured? How many biological replicates? What statistical tests were performed? What were the p values of the insignificant differences?

We appreciate the reviewer’s observations, prompting us to reanalyse the root hair experiments under low and high P condntions. In our earlier manuscript, we measured the length of root hairs originating from a zone spanning the end of the elongation zone into the differentiation zone. However, as root hairs are still elongating in this zone, we appreciate that this did not accurately represent the final length reached under different P conditions. In the revised manuscript, we only measured root hairs that had fully elongated in all genotypes and treatments (analysing 25 fully elongated root hairs from each root in at least 15 replicates). Moreover, we performed the root hair elongation in three independent biological replicates. Student T-test was used to calculate the test of significance. The bars represent standard error. P values have also been added to specific bars if they exhibited insignificant differences.

Secondly, although the authors claim that there are no significant differences in root hair lenght under HP conditions, a quick measurement of the bar lengths in panel C reveals that the root hair length increase when grown on LP is 3-fold for both wild type and *taa1*, and 2.5-fold for *aux1-22*. As I argued in my review of the manuscript by Giri et al., root hairs of all 3 genotypes respond to LP by increasing their root hair length ~3-fold. Thus, the scale of the root hair response to LP is similar in all 3 genotypes, suggesting that the mutated genes have no effect on P perception. One could use a similar line of reasoning for the *dao1.1* mutant: the authors are correct in claiming that the 10-fold increase in root hair length of wild type plants is reduced in *dao1.1* plants. However, the root hairs of *dao1.1* are so much longer than wild type root hairs under HP conditions, that, to increase 10-fold in length, they should reach lengths that are at least 2-fold longer than the maximum length that has been observed in wild type root hairs under any condition. This suggests that genetic factors controlling maximal root hair length, and not a decrease in *dao1* activity, are responsible for the reduced increase in root hair length.

Our reanalysed experiments revealed that root hair elongation is significantly disrupted in *aux1*, *taa1* mutants as compared to WT Columbia (which exhibited 2, 8 and 20 fold changes respectively). Hence *taa1* and *aux1* mutations attenuated low P induced root hair elongation 60 and 90%, respectively, compared to WT. We therefore reason that

both mutated genes have a major effect on this P response. Consistent with this, we observed that *taa1* and *aux1* mutations completely blocked low P induced root hair elongation in a buffered P system (Fig. S2). We also repeated and reanalysed high and low P root hair elongation assays in rice and observed that *osaux1* reduced this adaptive response over 90% compared to its WT control. Unlike, *aux1* and *taa1* mutants, *dao1.1* root hairs exhibit more elongation as compared to WT under both low and high P conditions, consistent with its role in auxin degradation.

In Fig 1e., part of the max. intensity projection appears to be missing. There appears to be selective DR5:NLS-Venus expression in median epidermal/cortical cell files, while expression is absent in the upper, more central cell files. In addition, the microscopy data would benefit from a quantitative analysis. I could argue that a slight difference in focal plane or root curvature between C and D is causing the decreased fluorescence in the centre of the roots. I would recommend the authors to quantify their data, provide the number of replicates and use internal controls (for example propidium iodide staining intensities) to correct for, for example, differences in working distance.

To help clarify, Figure 1E is not a maximum projection image. It is a single image on the root surface (surface view) which reveals a clear difference in DR5:VENUS fluorescence between HP and LP conditions. We reason that differences in intensity in panels C and D are due to different focal planes in these images. This is backed up by our maximum projection images (G and H) through the entire root that shows a clear difference between HP and LP.

The data in Fig. 4 and Fig. 5 should be quantified, but appears convincing and promising.

We appreciate the reviewer's suggestion. We have now quantified the fluorescence intensities of RSL2-GFP, RSL4-GFP and DR5-Venus reporters under low and high P condition and included in this manuscript (see Fig. 5 and S5).

Reviewers' comments:

Reviewer #1 (Remarks to the Author):

This new version of the manuscript is significantly improved respect to the original paper as new data and explanations of some results are now provided. The major concern regarding the root hair response to low Pi of the *ntaa1* and *aux1* mutants is apparently clarified with a new analysis of a different region of the root, which shows that the response in these mutants is significantly different, both quantitatively and qualitatively, from that of the WT. This would make this manuscript novel and worthy of publication. However, is the point of view of this reviewer that the data supporting the conclusion that auxin synthesis and transport play a critical role regulating the root hair adaptive response to low Pi are still not completely convincing. My major concerns are the following:

- 1) In most reports root hair length in high Pi is between 200-300 μm and in low Pi is between 600 and 900 μm , with an average increase of 3 fold between low and high Pi (i.e Bates and Lynch, 1996). If I recall well, in the original data presented in the previous version of this paper, the increase in on root hair length in response to low Pi was about 3-fold for the WT, as well as for the *aat1* and *aux1* mutants, being consistent with previous publications. However, in the data presented in this version of the paper (Figure 2B and C), the increase in root hair length in the WT in low Pi is now about 20-fold, which is unusually high compared with other publications. In fact, it looks like root hairs in high Pi practically do not elongate and are hardly visible in the images presented in Fig 1 and Fig S4. Perhaps this is due to the experimental conditions used, but the discrepancy with other reports needs to be explained.
- 2) In fact, the root length in the images presented in figure 1A, used for determining root lengths in high and low Pi, seems to be different from the root hair lengths observed figures 1G and H, in which the difference in root length between high and low Pi seedlings does not appear as dramatic as 20-fold.
- 3) In figure 3, it can be clearly seen that expression of AUX 1 in LRC and epidermal cells in the *aux1-22* mutant indeed restore root hair growth under low Pi conditions, but also under high Pi conditions. Suggesting that auxin is required for root hair elongation but not specifically for the low Pi response. Moreover, root hair lengths presented in figure 3 for high and low Pi seedlings seems also to be inconsistent with a 20-fold difference in root hair length.
- 4) The authors find by GC-MS/MS analysis that there are higher levels of auxin in the root hair zone in low Pi seedlings than in high Pi seedlings. However, this root zone, according to the images presented in Fig 1A, correspond to a zone in which fully mature root hairs are present. These results need to be explained, because auxin is required for root hair formation and root hair elongation, but probably has not further effect on fully elongated mature root hairs. What is the relevance of a higher auxin concentration in a root zone where root hairs are completely elongated, whereas no difference is found in the region where root hairs are being formed and rapidly elongate?
- 5) In figures 1C and D, the authors show that a DR5:VENUS reporter gene has a higher level of expression in the root tip of low Pi seedlings than in high Pi seedlings, which is inconsistent with the data on auxin quantification that shows no difference in auxin content between both types of seedlings. These results would suggest that auxin sensitivity rather than auxin concentration is altered by low Pi in the root tip and could influence root hair elongation, as previously show for the increase in lateral root formation (Perez-Torres, et al Plant Cell, 2008).
- 6) In figure S1 data is presented, suggesting that AAT1 expression is regulated by Pi availability. Changes in AAT1 expression are modest and no statistical analysis is presented to know whether these differences are statistically significant.

In summary, I find that the main issue with the interpretation of the data regarding the role of TAA1 and AUX1 in the root hair response to low Pi remains a major concern. Is auxin synthesis and

transport regulating the response or an effect on auxin sensitivity is what regulates the response? The finding that the expression of ARF9 is enhanced in low Pi seedlings and results obtained with the arf19-3 mutant would be consistent with an effect on auxin sensitivity. Perhaps using transfer experiment from high to low Pi would allow more direct comparison of root hair length and root hair elongation rates between high and low Pi seedlings in a similar root zone of the WT and auxin mutants. Since as proposed by the authors and other groups the effect of low Pi on root hair elongation is a local response, the effect should be easily measurable for the root hairs that are formed upon transfer to low Pi. If the difference in root hair length and root hair elongation rate between the WT and the mutants in response to low Pi is statistically different under these conditions, it would provide compelling evidence supporting the author's conclusions.

Response to reviewer 1 comments

Reviewer #1 (Remarks to the Author):

This new version of the manuscript is significantly improved respect to the original paper as new data and explanations of some results are now provided. The major concern regarding the root hair response to low Pi of the *ntaa1* and *aux1* mutants is apparently clarified with a new analysis of a different region of the root, which shows that the response in these mutants is significantly different, both quantitatively and qualitatively, from that of the WT. This would make this manuscript novel and worthy of publication. However, is the point of view of this reviewer that the data supporting the conclusion that auxin synthesis and transport play a critical role regulating the root hair adaptive response to low Pi are still not completely convincing. My major concerns are the following:

1) In most reports root hair length in high Pi is between 200-300 μM and in low Pi is between 600 and 900 μM , with an average increase of 3 fold between low and high Pi (i.e Bates and Lynch, 1996). If I recall well, in the original data presented in the previous version of this paper, the increase in on root hair length in response to low Pi was about 3-fold for the WT, as well as for the *aat1* and *aux1* mutants, being consistent with previous publications. However, in the data presented in this version of the paper (Figure 2B and C), the increase in root hair length in the WT in low Pi is now about 20-fold, which is unusually high compared with other publications. In fact, it looks like root hairs in high Pi practically do not elongate and are hardly visible in the images presented in Fig 1 and Fig S4. Perhaps this is due to the experimental conditions used, but the discrepancy with other reports needs to be explained.

We agree with the reviewer that the data presented in figure 2B and C shows a 20 fold increase in root hair elongation under low P compared to high P. This can be explained in part on the basis of our measurement criteria, where we only selected root hairs from the zone of highest root hair elongation. In addition, root hairs were considerably smaller in high P in our experimental conditions. Please note that we have also performed experiments using Bates and Lynch conditions (that employ a buffered P system) and data was presented in the original Figure S2. Under the latter experimental conditions, we find a 3 fold increase in root hair length in WT roots in LP versus HP conditions, but no difference in root hair length between HP and LP in *taa1* and *aux1* mutant backgrounds. These results clearly show that root hair elongation under low P conditions is TAA1 and AUX1 dependent.

2) In fact, the root length in the images presented in figure 1A, used for determining root lengths in high and low Pi, seems to be different from the root hair lengths observed figures 1G and H, in which the difference in root length between high and low Pi seedlings does not appear as dramatic as 20-fold.

The results presented in figure 1A and figure 1G and H are from two different experiments. Similarly, results presented in figure 2B and C are from another set of experiments. We do find variation in root hair phenotype from one experiment to the other and we also find that the root tip needs to be in contact with the growth medium on the plate. This appears to have some impact on the extent of root hair elongation and results in experiment to experiment, plate to plate and seedling to seedling variation.

3) In figure 3, it can be clearly seen that expression of AUX 1 in LRC and epidermal cells in the *aux1-22* mutant indeed restore root hair growth under low Pi conditions, but also under high Pi conditions. Suggesting that auxin is required for root hair elongation but not specifically for the low Pi response.

We agree with the reviewer that auxin is required for root hair elongation. AUX1 functions to facilitate this increase in auxin levels in the root hair zone by mobilising shootward auxin transport. Under low P there is an increase in root hair length in WT compared to *aux1* demonstrating the importance of AUX1 in low P mediated root hair elongation response.

Moreover, root hair lengths presented in figure 3 for high and low Pi seedlings seems also to be inconsistent with a 20-fold difference in root hair length. Please refer to our response to points 1 and 2.

4) The authors find by GC-MS/MS analysis that there are higher levels of auxin in the root hair zone in low Pi seedlings than in high Pi seedlings. However, this root zone, according to the images presented in Fig 1A, correspond to a zone in which fully mature root hairs are present. These results need to be explained, because auxin is required for root hair formation and root hair elongation, but probably has not further effect on fully elongated mature root hairs. What is the relevance of a higher auxin concentration in a root zone where root hairs are completely elongated, whereas no difference is found in the region where root hairs are being formed and rapidly elongate?

We apologise for not being clearer. The root hair zone used for mass spec analysis includes both immature root hairs and mature root hairs. Figure 1 H also clearly show a strong DR5 signal in the immature root hair zone, consistent with increased auxin accumulation/response in these cells. We have also provided more details in the methods section that has now been revised as below:

Auxin estimation

Roots tip (~1 mm) and root hair zone (next 1.5 mm region) from nine-days-old seedlings grown on low and high P media were cut using a sharp scissor and snap frozen in liquid nitrogen. 75 roots were collected per replicate with four replicates per experiment. Five-hundred picograms of $^{13}\text{C}_6$ -IAA internal standard was added to each sample before purification. Auxin estimation was done as described previously⁴⁶, with minor modifications.

5) In figures 1C and D, the authors show that a DR5:VENUS reporter gene has a higher level of expression in the root tip of low Pi seedlings than in high Pi seedlings, which is inconsistent with the data on auxin quantification that shows no difference in auxin content between both types of seedlings. These results would suggest that auxin sensitivity rather than auxin concentration is altered by low Pi in the root tip and could influence root hair elongation, as previously show for the increase in lateral root formation (Perez-Torres, et al Plant Cell, 2008).

Reporter based methods are more accurate for providing spatial resolution (i.e. tissue differences) compared to mass spec based methods where all tissues in a given samples are profiled. Therefore some differences between these techniques can be expected. Nevertheless, DR5 and direct auxin measurement by mass spectroscopy reveal an increase in auxin response and/or content in the root hair zone, consistent with our interpretation.

6) In figure S1 data is presented, suggesting that TAA1 expression is regulated by Pi availability. Changes in TAA1 expression are modest and no statistical analysis is presented to know whether these differences are statistically significant.

We can confirm that 24 hours after transfer to low P or high P media, GUS staining in TAA1-GUS roots was significantly higher in low P roots compared to high P. This has now been corrected in this revised version.

In summary, I find that the main issue with the interpretation of the data regarding the role of TAA1 and AUX1 in the root hair response to low Pi remains a major concern. Is auxin synthesis and transport regulating the response or an effect on auxin sensitivity is what regulates the response? The finding that the expression of ARF19 is enhanced in low Pi seedlings and results obtained with the arf19-3 mutant would be consistent with an effect on auxin sensitivity. Perhaps using transfer experiment from high to low Pi would allow more direct comparison of root hair length and root hair elongation rates between high and low Pi seedlings in a similar root zone of the WT and auxin mutants. Since as proposed by the authors and other groups the effect of low Pi on root hair elongation is a local response, the effect should be easily measurable for the root hairs that are formed upon transfer to low Pi. If the difference in root hair length and root hair elongation rate between the

WT and the mutants in response to low Pi is statistically different under these conditions, it would provide compelling evidence supporting the author's conclusions.

The auxin sensitivity model is an interesting idea (consistent with the *arf19* phenotype). However, it does not explain the root hair defect under low P in either auxin synthesis and transport mutant's *taa1* or *aux1* respectively. Hence, differences in level AND sensitivity are likely to be important.

To test the reviewer's point, we have performed an experiment to differentiate between changes in auxin levels versus sensitivity. Our mass spec data suggest that roots grown under low P have 2 fold more auxin compared to high P grown roots. Given this fact, we reasoned that roots grown under low P will be less sensitive to external auxin application (as their roots are already inhibited because of increased auxin accumulation under low P). We therefore grew wild type seedlings on low or high P and then transferred them to a range of 1-NAA concentrations (0 to 250nM). Any change in root elongation was scored 24 hours after transfer.

1-NAA treatment to LowP and HighP treated seedlings

The data presented in the figure above show

1. Low P roots grew slower than that of high P roots in control plates (0 nM 1-NAA) and after 24 hours they were shorter by ~ 50% compared to high P roots. This is consistent with our mass spec findings which detected ~2 fold increase in root auxin levels in low P conditions. This argues against an increase in auxin sensitivity because in that case the inhibition of root growth would have been more pronounced.
2. The low P roots are less sensitive to inhibition by an increasing concentration of 1-NAA as they are already inhibited by virtue of higher endogenous auxin levels under low P.
3. At very high auxin concentrations (150 -250nM 1-NAA), both the low P and high P roots show a similar inhibition of root growth presumably due to reaching a threshold level.

Taken together, experiments support that low P results in increased auxin levels through the activity of TAA1 and this auxin is transported to the root hair zone via AUX1. Our proposed model clearly explains the roles of TAA1 and AUX1 and is consistent with the root hair

elongation defect under low P we see in *taa1* and *aux1* mutants. This model also explains why low P mediated root hair elongation is dependent on auxin inducible ARF19 but not ARF7 which is not auxin inducible. Our model provides a mechanistic framework for root hair elongation under low P (please see schematic below now added as new figure 6).

Fig. 6. Model for root hair elongation under low P.

Low P elevates IAA levels in the root hair zone facilitated by TAA1 mediated auxin synthesis and AUX1 dependent auxin transport. The resulting increase in IAA levels leads to the induction of ARF19 which targets RSL2 and RSL4 and other downstream genes to promote root hair elongation.

REVIEWERS' COMMENTS:

Reviewer #4 (Remarks to the Author):

This manuscript propose a molecular mechanism by which low-phosphate in the media induces auxin biosynthesis, auxin transport by AUX1, auxin signalling triggered by ARF19 and downstream expression of RSL2 and RSL4 transcription factors. This mechanism is based on experiments done in mutants (*taa1*, *dao1/2*, *arf19*), auxin reporter (*DR5*), ARF reporters (*ARF19*) and TF reporters (*RSL2* and *RSL4*). All shown data fit into the proposed model although some direct connections between these components remain to be shown in contradiction with the title "Mechanistic framework.....".

Major concerns:

1. The lack of evidence on the binding of ARF19 on RSL4 promoter and also on RSL2 promoter under low-Pi. A ChIP experiment could prove these connections easily.

The presence of several Aux-RE sites on RSL4, the binding of over-expressed ARF5 on RSL4 by ChIP and the positive regulation by several ARFs (e.g. ARF5,7,8) on the expression of RSL4 was shown previously in Mangano et al (2017) although the growth conditions were slightly different.

Are there any Aux-RE sites in RSL2 promoter?. The authors should analyze this aspect since *rsl2* mutants showed a much profound phenotype than *rsl4* mutant in low-Pi and most of the paper is focused on RSL4 instead of RSL2. Which is the relationship between RSL2 and RSL4 under low Pi?. Is the double mutant *rsl2-rsl4* able to respond to low-Pi?.

2. How low-Pi triggers the expression of TAA1 in the root apex and the expression of AUX1 in the epidermis cells?. There is no mechanism proposed for this. Has PIN2 any importance on this?. Previously it was shown that AUX1 and PIN2 were required to transport IAA on the epidermis cells (especially atricoblast) of the root (by Claire Grierson NCB). How DAO1 up-regulation (that triggers auxin degradation?) can be explained under low-Pi where auxin biosynthesis and auxin signaling is going up?. The mutant *dao1* (possibly with higher auxin levels) showed longer root hairs under low-Pi.

3. These *arf19* mutants lack ARF19 expression?. I don't see the real time PCR characterization of ARF19 expression in these new isolates of *arf19* mutants or a reference if these *arf19* mutants have been characterized before.

Reviewer #5 (Remarks to the Author):

I was asked to comment specifically on the response to Reviewer 1.

My comments are in Green Italics on the below copy of the response.

Response to reviewer 1 comments

Reviewer #1 (Remarks to the Author):

This new version of the manuscript is significantly improved respect to the original paper as new data and explanations of some results are now provided. The major concern regarding the root hair response to low Pi of the *ntaa1* and *aux1* mutants is apparently clarified with a new analysis of a different region of the root, which shows that the response in these mutants is significantly different, both quantitatively and qualitatively, from that of the WT. This would make this manuscript novel and worthy of publication. However, is the point of view of this reviewer that the data supporting the conclusion that auxin synthesis and transport play a critical role regulating the root hair adaptive response to low Pi are still not completely convincing. My major concerns are the following:

1) In most reports root hair length in high Pi is between 200-300 μM and in low Pi is between 600 and 900 μM , with an average increase of 3 fold between low and high Pi (i.e Bates and Lynch, 1996). If I recall well, in the original data presented in the previous version of this paper, the increase in on root hair length in response to low Pi was about 3-fold for the WT, as well as for the *aat1* and *aux1* mutants, being consistent with previous publications. However, in the data presented in this version of the paper (Figure 2B and C), the increase in root hair length in the WT in low Pi is now about 20-fold, which is unusually high compared with other publications. In fact, it looks like root hairs in high Pi practically do not elongate and are hardly visible in the images presented in Fig 1 and Fig S4. Perhaps this is due to the experimental conditions used, but the discrepancy with other reports needs to be explained.

We agree with the reviewer that the data presented in figure 2B and C shows a 20 fold increase in root hair elongation under low P compared to high P. This can be explained in part on the basis of our measurement criteria, where we only selected root hairs from the zone of highest root hair elongation. In addition, root hairs were considerably smaller in high P in our experimental conditions. Please note that we have also performed experiments using Bates and Lynch conditions (that employ a buffered P system) and data was presented in the original Figure S2. Under the latter experimental conditions, we find a 3 fold increase in root hair length in WT roots in LP versus HP conditions, but no difference in root hair length between HP and LP in *taa1* and *aux1* mutant backgrounds. These results clearly show that root hair elongation under low P conditions is TAA1 and AUX1 dependent.

I agree with the authors here. Root phenotypes are extremely sensitive to environmental conditions that are difficult to control, including humidity, moisture levels and at least 36 different nutrients. It is very difficult to achieve completely identical values for all of these in every re-run of an experiment. Consequently, quantitative reproducibility between repeats of root experiments is not usually achieved. Instead conclusions are typically drawn from qualitative differences.

2) In fact, the root length in the images presented in figure 1A, used for determining root lengths in high and low Pi, seems to be different from the root hair lengths observed figures 1G and H, in which the difference in root length between high and low Pi seedlings does not appear as dramatic as 20-fold.

The results presented in figure 1A and figure 1G and H are from two different experiments. Similarly, results presented in figure 2B and C are from another set of experiments. We do find variation in root hair phenotype from one experiment to the other and we also find that the root tip needs to be in contact with the growth medium on the plate. This appears to have some impact on the extent of root hair elongation and results in experiment to experiment, plate to plate and seedling to seedling variation.

See 1.

3) In figure 3, it can be clearly seen that expression of AUX 1 in LRC and epidermal cells in the aux1-22 mutant indeed restore root hair growth under low Pi conditions, but also under high Pi conditions. Suggesting that auxin is required for root hair elongation but not specifically for the low Pi response.

We agree with the reviewer that auxin is required for root hair elongation. AUX1 functions to facilitate this increase in auxin levels in the root hair zone by mobilising shootward auxin transport. Under low P there is an increase in root hair length in WT compared to aux1 demonstrating the importance of AUX1 in low P mediated root hair elongation response.

I agree with the authors.

Moreover, root hair lengths presented in figure 3 for high and low Pi seedlings seems also to be inconsistent with a 20-fold difference in root hair length. Please refer to our response to points 1 and 2.

4) The authors find by GC-MS/MS analysis that there are higher levels of auxin in the root hair zone in low Pi seedlings than in high Pi seedlings. However, this root zone, according to the images presented in Fig 1A, correspond to a zone in which fully mature root hairs are present. These results need to be explained, because auxin is required for root hair formation and root hair elongation, but probably has not further effect on fully elongated mature root hairs. What is the relevance of a higher auxin concentration in a root zone where root hairs are completely elongated, whereas no difference is found in the region where root hairs are being formed and rapidly elongate?

We apologise for not being clearer. The root hair zone used for mass spec analysis includes both immature root hairs and mature root hairs. Figure 1 H also clearly show a strong DR5 signal in the immature root hair zone, consistent with increased auxin accumulation/response in these cells. We have also provided more details in the methods section that has now been revised as below:

Auxin estimation

Roots tip (~1 mm) and root hair zone (next 1.5 mm region) from nine-days-old seedlings grown on low and high P media were cut using a sharp scissor and snap frozen in liquid nitrogen. 75 roots were collected per replicate with four replicates per experiment. Five-hundred picograms of ¹³C₆-IAA internal standard was added to each sample before purification. Auxin estimation was done as described previously⁴⁶, with minor modifications.

I accept this response.

5) In figures 1C and D, the authors show that a DR5:VENUS reporter gene has a higher level of expression in the root tip of low Pi seedlings than in high Pi seedlings, which is inconsistent with the data on auxin quantification that shows no difference in auxin content between both types of seedlings. These results would suggest that auxin sensitivity rather than auxin concentration is altered by low Pi in the root tip and could influence root hair elongation, as previously show for the increase in lateral root formation (Perez-Torres, et al Plant Cell, 2008).

Reporter based methods are more accurate for providing spatial resolution (i.e. tissue differences) compared to mass spec based methods where all tissues in a given samples are profiled. Therefore some differences between these techniques can be expected. Nevertheless, DR5 and direct auxin measurement by mass spectroscopy reveal an increase in auxin response and/or content in the root hair zone, consistent with our interpretation.

I agree with the authors that their results suggest that both auxin levels and sensitivity are involved in the response of root hair length to P.

6) In figure S1 data is presented, suggesting that TAA1 expression is regulated by Pi availability. Changes in

TAA1 expression are modest and no statistical analysis is presented to know whether these differences are statistically significant.

We can confirm that 24 hours after transfer to low P or high P media, GUS staining in TAA1-GUS roots was significantly higher in low P roots compared to high P. This has now been corrected in this revised version.

It looks from figure S1 as if only the results in figure S1A have statistical support? The results in S1B are suggestive but must be considered preliminary if they are not statistically supported.

In summary, I find that the main issue with the interpretation of the data regarding the role of TAA1 and AUX1 in the root hair response to low Pi remains a major concern. Is auxin synthesis and transport regulating the response or an effect on auxin sensitivity is what regulates the response? The finding that the expression of ARF19 is enhanced in low Pi seedlings and results obtained with the *arf19-3* mutant would be consistent with an effect on auxin sensitivity. Perhaps using transfer experiment from high to low Pi would allow more direct comparison of root hair length and root hair elongation rates between high and low Pi seedlings in a similar root zone of the WT and auxin mutants. Since as proposed by the authors and other groups the effect of low Pi on root hair elongation is a local response, the effect should be easily measurable for the root hairs that are formed upon transfer to low Pi. If the difference in root hair length and root hair elongation rate between the WT and the mutants in response to low Pi is statistically different under these conditions, it would provide compelling evidence supporting the author's conclusions.

The auxin sensitivity model is an interesting idea (consistent with the *arf19* phenotype). However, it does not explain the root hair defect under low P in either auxin synthesis and transport mutant's *taa1* or *aux1* respectively. Hence, differences in level AND sensitivity are likely to be important.

I agree with the authors.

To test the reviewer's point, we have performed an experiment to differentiate between changes in auxin levels versus sensitivity. Our mass spec data suggest that roots grown under low P have 2 fold more auxin compared to high P grown roots. Given this fact, we reasoned that roots grown under low P will be less sensitive to external auxin application (as their roots are already inhibited because of increased auxin accumulation under low P). We therefore grew wild type seedlings on low or high P and then transferred them to a range of 1-NAA concentrations (0 to 250nM). Any change in root elongation was scored 24 hours after transfer.

1-NAA treatment to LowP and HighP treated seedlings

— HighP — LowP

The data presented in the figure above show

1. Low P roots grew slower than that of high P roots in control plates (0 nM 1-NAA) and after 24 hours they were shorter by ~ 50% compared to high P roots. This is consistent with our mass spec findings which detected ~2 fold increase in root auxin levels in low P conditions. This argues against an increase in auxin sensitivity because in that case the inhibition of root growth would have been more pronounced.
2. The low P roots are less sensitive to inhibition by an increasing concentration of 1-NAA as they are already inhibited by virtue of higher endogenous auxin levels under low P.
3. At very high auxin concentrations (150 -250nM 1-NAA), both the low P and high P roots show a similar inhibition of root growth presumably due to reaching a threshold level.

This graph helps a bit with the arguments but not very much. It would not significantly strengthen the paper and there is no need to include it.

Taken together, experiments support that low P results in increased auxin levels through the activity of TAA1 and this auxin is transported to the root hair zone via AUX1. Our proposed model clearly explains the roles of TAA1 and AUX1 and is consistent with the root hair elongation defect under low P we see in *taa1* and *aux1* mutants. This model also explains why low P mediated root hair elongation is dependent on auxin inducible ARF19 but not ARF7 which is not auxin inducible. Our model provides a mechanistic framework for root hair elongation under low P (please see schematic below now added as new figure 6).

Fig. 6. Model for root hair elongation under low P.

Low P elevates IAA levels in the root hair zone facilitated by TAA1 mediated auxin synthesis and AUX1 dependent auxin transport. The resulting increase in IAA levels leads to the induction of ARF19 which targets RSL2 and RSL4 and other downstream genes to promote root hair elongation.

Whilst I appreciate the conceptual benefits of including a model like this, this version is oversimplified and potentially misleading. The expression data in Figures 5C and 4E suggest that ARF19 is expressed in all zones of the root (tip, elongation zone, differentiation zone), RSL2 in the differentiation zone, and RSL4 in the elongation zone. The authors present no evidence in this manuscript that these three genes all act simultaneously in epidermal cells emerging from beneath the lateral root cap as shown in this diagram. Also, the order in which root hair cells express these genes appears from the images in the figures to be: ARF19, which possibly remains expressed throughout root hair development then RSL2 in the differentiation zone only, then RSL4 in the elongation zone. It is likely some combination of the effects of these factors with each other and/or with other factors as yet unidentified that produces the effect on hair length. This diagram and its legend need to be altered to make these nuances clear.

As requested, please see below our point-by-point response to issues raised by the referees.

Reviewer #4 (Remarks to the Author):

This manuscript propose a molecular mechanism by which low-phosphate in the media induces auxin biosynthesis, auxin transport by AUX1, auxin signalling triggered by ARF19 and downstream expression of RSL2 and RSL4 transcription factors. This mechanism is based on experiments done in mutants (*taa1*, *dao1/2*, *arf19*), auxin reporter (*DR5*), ARF reporters (*ARF19*) and TF reporters (*RSL2* and *RSL4*). All shown data fit into the proposed model although some direct connections between these components remain to be shown in contradiction with the title "Mechanistic framework.....".

Major concerns:

1. The lack of evidence on the binding of ARF19 on RSL4 promoter and also on RSL2 promoter under low-Pi. A ChIP experiment could prove these connections easily. The presence of several Aux-RE sites on RSL4, the binding of over-expressed ARF5 on RSL4 by ChIP and the positive regulation by several ARFs (e.g. ARF5,7,8) on the expression of RSL4 was shown previously in Mangano et al (2017) although the growth conditions were slightly different.

[Editorial Note: Unpublished data redacted by Editorial Team as per Authorial Request.]

Are there any Aux-RE sites in RSL2 promoter?. The authors should analyze this aspect since *rsl2* mutants showed a much profound phenotype than *rsl4* mutant in low-Pi and most of the paper is focused on RSL4 instead of RSL2. Which is the relationship between RSL2 and RSL4 under low Pi?. Is the double mutant *rsl2-rsl4* able to respond to low-Pi?.

Our response:

As suggested by the reviewer and editor, a revised schematic showing Aux-RE's in the RSL2 promoter has now been included in Supplementary Figure 6 and text in the main manuscript has been modified as follows: *RSL4* has been shown to be induced by auxin¹⁹. We searched for the presence of auxin response element(s) in the *RSL4* and *RSL2* promoter sequence. Motif analysis revealed highly conserved auxin response elements in both *RSL* promoters (Supplementary Figure 6b) that represent target(s) for ARFs such as ARF19.

An initial characterisation of the *rs/2 rs/4* double mutant revealed that its root hair phenotype under low P conditions was more severe than the single mutants.

2. How low-Pi triggers the expression of TAA1 in the root apex and the expression of AUX1 in the epidermis cells?. There is no mechanism proposed for this. Has PIN2 any importance on this?. Previously it was shown that AUX1 and PIN2 were required to transport IAA on the epidermis cells (especially atricoblast) of the root (by Claire Grierson NCB).

Our response:

To clarify, we show that there is an increase in the *TAA1* expression in the root apex under low P

(but we have not shown an increase in *AUX1* expression in the epidermis under low P). However, it is not clear at present how low P results in an increase in *TAA1* expression in the root apex. Nevertheless, our *TAA1* expression and mutant data helps explain the basis for increased IAA levels under low P conditions.

Auxin efflux carrier PIN2 is expressed in the lateral root cap cells and in the epidermal and cortical cells and is the key PIN protein facilitating shootward movement of auxin. (Muller et al, 1998, *EMBO Journal* 17: 6903–6911). Mutation in either AUX1 or PIN2 result in root hair elongation defect as both *aux1* and *pin2* mutants have reduced root hair length and root hair density (Rigas et al, 2013, *New Phytologist* 197: 1130–1141). It has also been reported that *pin2* mutants have reduced root hair density under low P compared to wildtype Columbia seedlings (Kumar et al 2015, *Journal of Experimental Botany*, 66, 1499–1510).

How DAO1 up-regulation (that triggers auxin degradation?) can be explained under low-Pi where auxin biosynthesis and auxin signaling is going up?. The mutant *dao1* (possibly with higher auxin levels) showed longer root hairs under low-Pi.

Our response:

We have previously shown (in Porco et al, 2016, PNAS) that AtDAO1 encodes an IAA oxidase, hence *DAO1* mRNA up-regulation would be expected to result in increased auxin breakdown. In the two *DAO* alleles we used (*dao1.1* and *dao1.2D*), we only observed a low P root hair elongation defect in the gain of function *dao1.2D* allele

(which has higher *DAO1* expression and lower levels of IAA). This makes sense since the *dao1.2D* (but not the loss of function *dao1.1*) allele will impact high auxin phenotypes such as root hair elongation under low P.

3. These *arf19* mutants lack ARF19 expression?. I don't see the real time PCR characterization of ARF19 expression in these new isolates of *arf19* mutants or a reference if these *arf19* mutants have been characterized before.

Our response:

Our RT-PCR experiments revealed that no *ARF19* transcript was detected in either the *arf19-a* and *arf19-b* alleles, confirming that they represent null alleles (consistent with T-DNA insertions within their coding sequences). In contrast, in the *arf19-3* allele we did detect the transcript, which could possibly be explained by its T-DNA insertion being 8 bases upstream of the transcription start site. Irrespective, as we cannot guarantee that *arf19-3* represents a null allele, we have removed data relating to *arf19-3* from the manuscript.

Reviewer #5 (Remarks to the Author):

Our response:

We are very pleased to note that this reviewer is in general agreement with our response to the original reviewer 1 comments and has not raised any major concerns. However, the reviewer pointed out that the model presented in Figure 6 was too simplistic. The reviewer commented:

Whilst I appreciate the conceptual benefits of including a model like this, this version is oversimplified and potentially misleading. The expression data in Figures 5C and 4E suggest that ARF19 is expressed in all zones of the root (tip, elongation zone, differentiation zone), RSL2 in the differentiation zone, and RSL4 in the elongation zone. The authors present no evidence in this manuscript that these three genes all act simultaneously in epidermal cells emerging from beneath the lateral root cap as shown in this diagram. Also, the order in which

root hair cells express these genes appears from the images in the figures to be: ARF19, which possibly remains expressed throughout root hair development then RSL2 in the differentiation zone only, then RSL4 in the elongation zone. It is likely some combination of the effects of these factors with each other and/or with other factors as yet unidentified that produces the effect on hair length. This diagram and its legend need to be altered to make these nuances clear.

We completely agree with the reviewer that our model was too simplistic. We have now modified this figure and the figure legend (and the text in the manuscript file) revised as below:

Model for root hair elongation under low P

Low P elevates IAA levels in the root apex facilitated by TAA1 mediated auxin synthesis and AUX1 dependent auxin transport. The resulting increase in IAA levels leads to the induction of ARF19 in the root apex resulting in RSL2 and RSL4 induction in the elongation and differentiation zones respectively promoting root hair elongation.